# Unveiling the wealth-bone connection: How socioeconomic status influences trabecular bone health

Yuan Zhang[1☯], Tingxiao Zhao[2☯], Yanlei Li[3], Kai Chen[1*], Qice Sun[1*]

**1** Department of Rheumatology, The Second Affiliated Hospital of Zhejiang Chinese Medical University, Hangzhou, China, **2** Center for Plastic and Reconstructive Surgery, Department of Orthopedics, Zhejiang Provincial People's Hospital (Affiliated People's Hospital, Hangzhou Medical College), Hangzhou, Zhejiang, China, **3** Emergency and Critical Care Center, Department of Emergency, Zhejiang Provincial People's Hospital (Affiliated People's Hospital, Hangzhou Medical College), Hangzhou, Zhejiang, China

☯ These authors contribute equally to this work.
* xinhuasunqice@163.com (QS); drck2012@126.com (KC)

## Abstract

### Background

Currently, the relationship between socioeconomic status (SES) and trabecular bone score (TBS) remains unclear. This study uses existing data from the National Health and Nutrition Examination Survey (NHANES) to test the hypothesis that SES, measured by the poverty income ratio (PIR), is associated with TBS among adults in the United States of America.

### Methods

This study analyzed the relationship between SES and TBS in 7,832 adults using data from the 2005–2008 NHANES. The TBS data were obtained from pre-existing lumbar spine DXA scans collected in the NHANES database. The study employed a multivariable linear regression model, adjusting for various confounding factors, to examine the association between the PIR and TBS. Additionally, demographic characteristics were described, stratified analyses, single factor analysis, smooth curve fitting, and interaction analysis were conducted.

### Results

Results showed that PIR was positively correlated with TBS (β = 0.0019, 95% CI 0.0002–0.0035, P < 0.05), and high-income individuals had a TBS 0.018 higher than low-income individuals (95% CI 0.0106–0.0254, P < 0.001). Smooth curve fitting analysis revealed a linear relationship between PIR and TBS. Stratified analysis showed that this positive correlation was particularly evident among adults aged 50 and older, observed specifically in Mexican Americans and individuals of European and African ancestry.

**Data availability statement:** This study utilized the publicly available dataset from the NHANES in the United States (http://www.cdc.gov/nchs/nhanes/). The data were anonymized prior to analysis, participants provided informed consent, and approval was obtained from the Institutional Review Board of the National Center for Health Statistics.

**Funding:** The author(s) received no specific funding for this work.

**Competing interests:** The authors have declared that no competing interests exist.

## Conclusions

Our research shows that there is a significant positive correlation between adult SES and TBS. Higher household income levels correspond to better TBS levels, which can help prevent osteoporosis and fractures. When developing strategies for the prevention and treatment of osteoporosis, healthcare professionals and policymakers should consider the impact of socioeconomic inequalities.

## Introduction

Osteoporosis is a systemic metabolic bone disease characterized by decreased bone density and deterioration of bone microstructure, which often leads to increased bone fragility and a higher risk of fractures [1,2]. A population-based study found that among Americans aged 50 and older, 2.6 million had hip fractures, 14.6 million had wrist fractures, and 5.2 million had vertebral fractures [3]. As the population ages, fracture prevalence continues to rise. Currently, an estimated 14 million Americans and over 200 million people worldwide have osteoporosis [4,5]. Osteoporosis diagnosis traditionally relies on bone mineral density (BMD) measurements, with hip and vertebral fractures being the most severe outcomes. The Trabecular Bone Score (TBS) is a new FDA-approved tool that assesses bone microarchitecture using pixel greyscale analysis of lumbar dual-energy X-ray absorptiometry (DXA) scans [6]. TBS evaluates bone strength and predicts fracture risk partly independently of BMD and clinical risk factors, which can potentially change with pharmacological therapy [7–9]. A high TBS indicates strong, fracture-resistant bone microarchitecture, while a low TBS suggests weaker, more fracture-prone bone.

Socioeconomic status (SES) encompasses income, education, employment, and neighborhood socioeconomic characteristics [10]. Among various measures of poverty, household income relative to the federal poverty line more accurately reflects economic conditions. SES has a significant impact on the incidence of osteoporosis, but this factor is often overlooked [11–13]. A study in Spain found that poverty is a clear risk factor for osteoporotic fractures [14]; a recent cross-sectional study further confirmed the importance of SES in osteoporosis management [13]. Research in Louisiana showed that SES is positively correlated with BMD in the general population [15]. However, a 2009 systematic review found conflicting evidence regarding the association of osteoporotic fractures with income and educational level [16]. This inconsistency may be related to study methodologies or the limitation of using BMD as the sole indicator. Although BMD reflects bone density, it does not assess the microarchitecture of trabecular bone, which is equally important for fracture risk. The TBS can independently assess trabecular bone structure, and may therefore be a more effective indicator for osteoporosis and fracture risk than BMD. While many studies have explored the impact of SES on BMD, the effect of SES on TBS remains unclear.

This study aims to address the gap in recent research on the relationship between family poverty income ratio (PIR) and TBS, as well as osteoporosis risk in the general population. We hypothesize that higher PIR levels may be positively associated with

TBS. This study uses nationally representative sample data to examine the relationship between SES and TBS among adults in the United States. The results of this cross-sectional study will help to elucidate the association between SES and TBS, as well as the risk of osteoporosis, providing new insights for prevention strategies.

## Methods

### Study population

The NHANES is a nationally representative survey conducted by the National Center for Health Statistics (NCHS). It used a stratified, multistage probability cluster sampling design to select representative samples from the United States civilian population and assess their health or nutritional status [17]. The detailed data from NHANES are publicly available and have been approved by the Ethics Review Board of the NCHS in the United States; therefore, this study did not require separate ethical approval. The survey utilized the 2005–2008 NHANES dataset, focusing on participants aged over 20 who completed the family income interview and TBS assessment. Excluding those with incomplete family PIR, TBS, and covariate data, it included 7,832 participants. The specific process is shown in Fig 1.

### Study variables

We used the family PIR [18] as a measure of participant financial status, calculated by adjusting household income for annual changes in household size, cost of living, and the consumer price index relative to federally determined poverty thresholds. We divided the family PIR into three levels: low-income level (PIR ≤ 1.49), middle-income level (1.5 ≤ PIR ≤ 3.51), and high-income level (3.52 ≤ PIR ≤ 5).

TBS quantifies the three-dimensional microstructure of trabecular bone by analyzing pixel grayscale gradients and calculating local variations in lumbar spine DXA images [6]. The spine scans, which were obtained as part of the NHANES survey, utilized Hologic QDR-4500A fan-beam densitometers (Hologic, Inc., Bedford, Massachusetts). Subsequently, the total TBS score was estimated in adults aged 20 years and older using TBS software (Med-Imap SA TBS Calculator version 2.1.0.2).

Based on prior research [16] and clinical experience, we included the following covariates that might influence the association between family PIR and total TBS: age, gender, ethnicity, body mass index (BMI), alkaline phosphatase, total cholesterol, low-density lipoprotein cholesterol (LDL-C), high-density lipoprotein cholesterol (HDL-C), total protein, serum phosphorus, blood urea nitrogen, serum calcium, serum uric acid, serum cotinine, serum cadmium, serum lead, serum mercury, vitamin D (Vit D), C-reactive protein (CRP), cardiovascular disease, cancer status, and smoking status. Details of each variable are available on the NHANES website (http://www.cdc.gov/nchs/nhanes.htm).

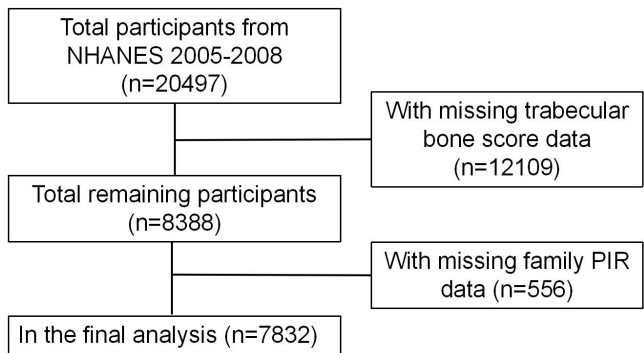

**Fig 1. Participant screening flowchart.**

All analyses in this study adopted the racial and ethnic categories defined by NHANES. For presentation in this manuscript, we use ancestry-based terminology to describe these populations: the "Non-Hispanic White" group is described as the "European ancestry" group, the "Non-Hispanic Black" group is described as the "African ancestry" group, and "Mexican American" is retained. We emphasize that these categories represent self-identified social groups and do not denote precise genetic ancestry.

## Statistical analysis

Our analyses accounted for the complexity of the NHANES survey design by incorporating sampling weights, primary sampling units, and strata. Descriptive statistics were used to summarize continuous and categorical variables. The weighted mean and standard error (SE) were used for continuous variables, while categorical variables were expressed as weighted percentages. Comparison between groups using weighted chi-square test and weighted linear regression. This study employs a weighted multivariate linear regression model to analyze the relationship between PIR and TBS, categorizing PIR into three levels for trend testing. To control for confounding factors, we established three progressively adjusted models: an unadjusted model, a demographically adjusted model, and a fully adjusted model. Subgroup analyses and interaction tests by age, sex, and ethnicity were also performed to assess the stability of the association across different populations. Based on the fully adjusted model, a generalized additive model (GAM) and smooth curve fitting were applied to examine the nature of the relationship between PIR and TBS. For nonlinear relationships, a two-stage linear regression and recursive algorithm were used to determine the inflection point between PIR and TBS.

A p-value less than 0.05 (two-sided) was considered statistically significant. All statistical analyses were performed using EmpowerStats (http://www.empowerstats.com) and R software (http://www.R-project.org).

## Results

### Baseline characteristics of study participants

This study included a total of 7,832 people, with 3,847 women and 3,985 men. Table 1 shows the demographic and clinical characteristics weighted by PIR quartiles. Significant differences were observed in the baseline characteristics of the PIR quartiles, except for the serum phosphorus, serum uric acid, BMI, LDL-C, and cardiovascular diseases. Notably, participants in the lower PIR household level were younger, had a higher proportion of women, and were more likely to belong to European ancestry groups. They had insufficient calcium intake, lower HDL-C and total cholesterol, were more prone to vitamin D deficiency, and were more likely to smoke or be exposed to smoking than those in the high PIR group.

### Associations between family PIR and total TBS

Table 2 shows the regression analysis results of family PIR and TBS. In the unadjusted model, PIR was significantly positively correlated with TBS (β = 0.0046, 95% CI: 0.0026–0.0066, P < 0.001). After adjusting for covariates, models 2 and 3 still maintained this significant correlation. In the fully adjusted model (model 3), for each unit increase in PIR, TBS increased by 0.0019 (95% CI: −0.0273–0.0222, P < 0.05). When grouped by income level, high-income families had a TBS higher by 0.018 compared to low-income families. Meanwhile, the trend test showed a P-value of P < 0.001, indicating that TBS significantly increases as family PIR levels rise. Moreover, in the generalized additive model and smoothing curve fitting, we found a linear relationship between PIR and TBS (Fig 2).

### Stratified analyses based on additional variables

Table 3 and Fig 3 show the results of stratified analysis by age, gender, and ethnicity. In the unadjusted model, all age groups showed a significant positive correlation between household PIR and TBS. However, after full adjustment, only the ≥ 50 years group maintained significance. Regarding gender, both males and females showed significant positive

 

**Table 1. Characteristics of the study participants according to PIR levels.**

| Characteristic | Q1 | Q2 | Q3 | Q4 | P value |
|---|---|---|---|---|---|
| Age(years) | 42.73±16.83 | 47.73±18.79 | 45.52±15.99 | 47.35±13.40 | <0.0001 |
| Gender (%) | | | | | 0.0025 |
| Male | 46.81 | 46.49 | 49.59 | 51.62 | |
| Female | 53.19 | 53.51 | 50.41 | 48.38 | |
| Race/ethnicity (%) | | | | | <0.0001 |
| Mexican American | 18.28 | 11.18 | 5.51 | 2.84 | |
| Other race/ethnicity | 12.82 | 11.44 | 8.87 | 7.58 | |
| Non-Hispanic white | 53.49 | 63.76 | 74.25 | 83.66 | |
| Non-Hispanic black | 15.42 | 13.61 | 11.37 | 5.93 | |
| Alkaline Phosphatase (U/L) | 72.32±24.65 | 69.76±23.92 | 65.86±23.42 | 65.20±21.28 | <0.0001 |
| Blood Urea Nitrogen (mg/dL) | 11.80±5.71 | 12.92±5.32 | 12.73±4.81 | 13.12±4.44 | <0.0001 |
| Serum Calcium (mg/dL) | 9.45±0.37 | 9.49±0.37 | 9.47±0.35 | 9.45±0.35 | 0.0035 |
| Serum Phosphorus (mg/dL) | 3.80±0.58 | 3.77±0.59 | 3.79±0.54 | 3.80±0.56 | 0.4598 |
| Total Cholesterol (mg/dL) | 198.92±45.99 | 200.77±42.69 | 198.72±40.64 | 202.08±38.22 | 0.019 |
| HDL-C (mg/dL) | 50.12±15.64 | 52.99±16.34 | 52.71±15.52 | 54.18±15.78 | <0.0001 |
| LDL-C (mg/dL) | 118.70±40.69 | 118.67±37.66 | 117.25±36.03 | 118.90±34.08 | 0.4168 |
| Total Protein (g/L) | 7.18±0.47 | 7.15±0.45 | 7.14±0.44 | 7.06±0.43 | <0.0001 |
| Serum Uric Acid (mg/dL) | 5.37±1.41 | 5.49±1.42 | 5.47±1.39 | 5.43±1.32 | 0.0926 |
| Serum Cotinine (ng/mL) | 108.00±159.75 | 73.85±142.48 | 69.02±136.36 | 46.48±116.11 | <0.0001 |
| Serum Cadmium (µg/L) | 0.73±0.91 | 0.59±0.66 | 0.55±0.69 | 0.44±0.45 | <0.0001 |
| Serum Lead (µg/L) | 1.92±1.69 | 1.83±1.47 | 1.70±1.41 | 1.61±1.13 | <0.0001 |
| Serum Mercury (µg/L) | 1.20±1.73 | 1.40±1.91 | 1.54±2.02 | 2.22±2.56 | <0.0001 |
| CRP (mg/L) | 0.45±0.85 | 0.40±0.89 | 0.36±0.62 | 0.36±0.83 | 0.0016 |
| BMI (Kg/m2) | 28.11±6.38 | 28.24±5.79 | 28.18±5.92 | 27.96±5.46 | 0.373 |
| Vit D (nmol/L) | 58.71±23.17 | 61.31±23.74 | 64.31±23.64 | 67.56±22.72 | <0.0001 |
| TBS | 1.37±0.15 | 1.37±0.15 | 1.39±0.14 | 1.39±0.14 | <0.0001 |
| Cardiovascular Diseases(%) | | | | | 0.6385 |
| Yes | 2.97 | 3.33 | 2.68 | 2.74 | |
| No | 97.03 | 96.67 | 97.32 | 97.26 | |
| Cancer (%) | | | | | 0.0483 |
| Yes | 7.29 | 8.37 | 7.61 | 9.47 | |
| No | 92.71 | 91.63 | 92.39 | 90.53 | |
| Smoking status | | | | | <0.0001 |
| Every day | 33.75 | 23.01 | 20.98 | 12.2 | |
| Some days | 4.92 | 3.45 | 3.46 | 3.19 | |
| Not at all | 17.05 | 26.01 | 23.3 | 28.22 | |
| Missing | 44.28 | 47.54 | 52.25 | 56.39 | |

Data are expressed as weighted means±SE or percentages (%). Racial/ethnic categories as defined by NHANES. Abbreviation: SE, standard error; HDL-C, high-density lipoprotein cholesterol; LDL-C, low-density lipoprotein cholesterol; CRP, C-reactive protein; Vit D, vitamin D; BMI, body mass index; TBS, trabecular bone score.

correlations after full adjustment. In the analysis by ancestry, a significant association was observed only in the group of European ancestry in the unadjusted model. After full adjustment, this association extended to groups of Mexican and African ancestry, while other groups showed no significance. Other racial groups showed no statistical significance before or after adjustment.

**Table 2. Association between the PIR and the TBS.**

| Variable | Model 1 β (95% CI) | Model 2 β (95% CI) | Model 3 β (95% CI) |
|---|---|---|---|
| Family PIR per 1 increase | 0.0046 (0.0026, 0.0066)*** | 0.0064 (0.0046, 0.0082)*** | 0.0019 (0.0002, 0.0035)* |
| Low-income level (0–1.49) | Ref | Ref | Ref |
| Middle-income level (1.5–3.51) | −0.0058 (−0.0165, 0.0049) | 0.0140 (0.0045, 0.0236)** | 0.0099 (0.0021, 0.0176)* |
| High-income level (3.52–5) | 0.0145 (0.0044, 0.0246)** | 0.0238 (0.0147, 0.0328)*** | 0.0180 (0.0106, 0.0254)*** |
| P for trend | <0.001 | <0.001 | <0.001 |

Model 1: Unadjusted;

Model 2: Adjusted for age, gender, and race/ethnicity;

Model 3: Further adjusted for alkaline phosphatase, blood urea nitrogen, serum calcium, total cholesterol, serum phosphorus, serum uric acid, total protein, serum cotinine, HDL-C, LDL-C, BMI, VIT D, CRP, serum cadmium, serum lead, serum mercury, cardiovascular diseases, cancer, smoking status. P<0.05 presents significant difference, with *p<0.05, **p<0.01, ***p<0.001.

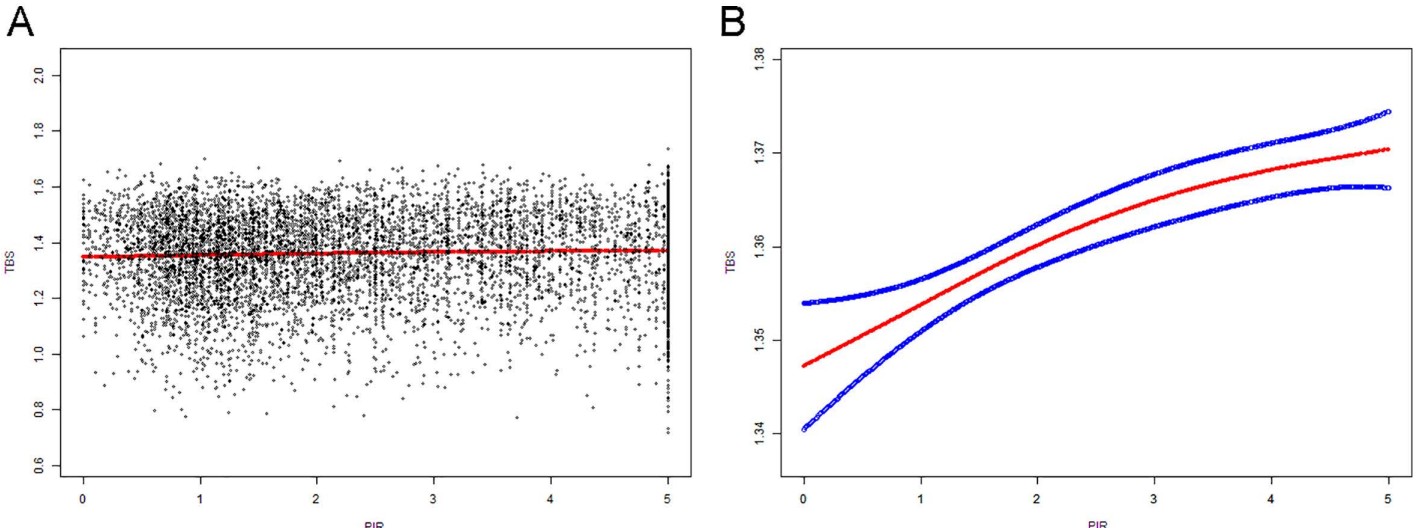

**Fig 2. The association between family PIR and total TBS. (A)** Each black point represents a sample. **(B)** The solid red line represents the smooth curve fit between variables. Blue bands represent the 95% of confidence interval from the fit. The model adjusts all variables except family PIR and total TBS.

## Discussion

In our study, we investigated the association between family PIR and total TBS among a large sample of the U.S. population over the age of 20. Our results showed that family PIR, whether as a continuous or categorical variable, was positively and linearly associated with total TBS when adjusted for potential confounding factors. The lower family income subgroup was younger, more often female, and characterized by a higher representation of individuals from European ancestry. They were also more likely to be daily smokers, suffer from vitamin D deficiency, and be exposed to environmental pollution more frequently than those in the higher family PIR category. After full adjustment, a significant association between family PIR and total TBS was found among participants aged ≥ 50, including both men and women, and specifically among Mexican Americans and individuals of European and African ancestry. This suggests that a higher family PIR is associated with a lower probability of osteoporosis and osteoporotic fractures.

**Table 3. Association of family PIR and TBS after stratification by age, gender, and race/ethnicity.**

| Subgroup | N | Model 1 β (95% CI) | Model 2 β (95% CI) | Model 3 β (95% CI) |
|---|---|---|---|---|
| Age | 7832 | | | |
| >=20, <35 | 1923 | 0.0065 (0.0034, 0.0096)*** | 0.0043 (0.0012, 0.0075)** | 0.0024 (−0.0004, 0.0051) |
| >=35, <50 | 2145 | 0.0089 (0.0054, 0.0124)*** | 0.0091 (0.0055, 0.0127)*** | 0.0018 (−0.0012, 0.0048) |
| >=50 | 3764 | 0.0123 (0.0095, 0.0152)*** | 0.0121 (0.0092, 0.0151)*** | 0.0104 (0.0078, 0.0129)*** |
| Total | 7832 | 0.0096 (0.0078, 0.0114)*** | 0.0091 (0.0072, 0.0109)*** | 0.0055 (0.0039, 0.0071)*** |
| Gender | | | | |
| Male | 3985 | −0.0002 (−0.0031, 0.0026) | 0.0033 (0.0006, 0.0060)* | 0.0050 (0.0028, 0.0071)*** |
| Female | 3847 | 0.0095 (0.0067, 0.0123)*** | 0.0091 (0.0066, 0.0116)*** | 0.0038 (0.0017, 0.0060)*** |
| Total | 7832 | 0.0047 (0.0027, 0.0068)*** | 0.0064 (0.0046, 0.0082)*** | 0.0037 (0.0021, 0.0052)*** |
| Race/ethnicity | | | | |
| Mexican American | 1426 | 0.0019 (−0.0031, 0.0069) | 0.0060 (0.0015, 0.0106)** | 0.0073 (0.0035, 0.0111)*** |
| Other race/ethnicity | 884 | −0.0009 (−0.0064, 0.0045) | 0.0033 (−0.0016, 0.0083) | 0.0003 (−0.0040, 0.0045) |
| Non-Hispanic white | 3862 | 0.0069 (0.0039, 0.0099)*** | 0.0073 (0.0047, 0.0099)*** | 0.0039 (0.0017, 0.0061)*** |
| Non-Hispanic black | 1660 | 0.0013 (−0.0033, 0.0060) | 0.0031 (−0.0012, 0.0073) | 0.0041 (0.0005, 0.0076)* |
| Total | 7832 | 0.0052 (0.0031, 0.0072)*** | 0.0064 (0.0046, 0.0082)*** | 0.0037 (0.0021, 0.0052)*** |

Model 1: unadjusted.

Model 2: adjusted for age, gender, and race/ethnicity.

Model 3: adjusted for all variables except family PIR, total TBS, and corresponding stratified variables.

P<0.05 presents significant difference, with *p<0.05, **p<0.01, ***p<0.001.

Racial/ethnic categories as defined by NHANES.

Although the effect size observed in this study was small, its clinical significance remains notable. As a continuous scale measurement, the range of TBS is relatively narrow, so even minor differences may influence the diagnosis of bone microarchitecture deterioration. Notably, this effect reflects the independent contribution of SES after adjusting for multiple factors, suggesting that the overall impact of SES on bone health may be more profound. From a population health perspective, subtle shifts in the distribution of risk factors can lead to substantial changes in disease burden. Given that SES affects a broad population, even small effects may accumulate to yield considerable public health implications for fracture prevention. The consistent associations observed across multiple subgroups in this study further affirm the clinical relevance of these findings.

Osteoporosis risk is partially determined by the cumulative effects of unhealthy dietary and activity behaviors occurring throughout life. TBS is a new complementary approach for osteoporosis evaluation in clinical practice. Several mechanisms may explain the relationship of family SES to bone health. First of all, it is likely that adults of low socioeconomic position may be at increased osteoporosis risk because of inadequate nutrition, calcium intake, and physical activity during childhood, resulting in suboptimal achievement of peak bone mass earlier in life [19,20]. Lower socioeconomic status, poverty, food insecurity, inadequate nutrition, and calcium plus vitamin D deficiency in older adulthood may result in accelerated loss of bone mass and accelerated development of osteoporosis with age [21,22]. Evidence from a cohort study involving 246 women with osteoporosis demonstrated that 25-hydroxyvitamin D (25OHD) was associated with spine TBS, affecting bone microarchitecture and mobility performance [23]. Secondly, higher income was associated with higher self-reported physical activity [24]. Economically active individuals may engage in regular exercise, which helps improve balance and postural stability, potentially resulting in a lower risk of osteoporosis [25,26]. A cohort study from Australia involving 894 men and 682 women (ages 24–98 years) showed that lower TBS was associated with older age, increased weight, lower childhood physical activity, and lower BMD in men, and with older age, shorter stature, lower BMD, and lower mobility in women [27].

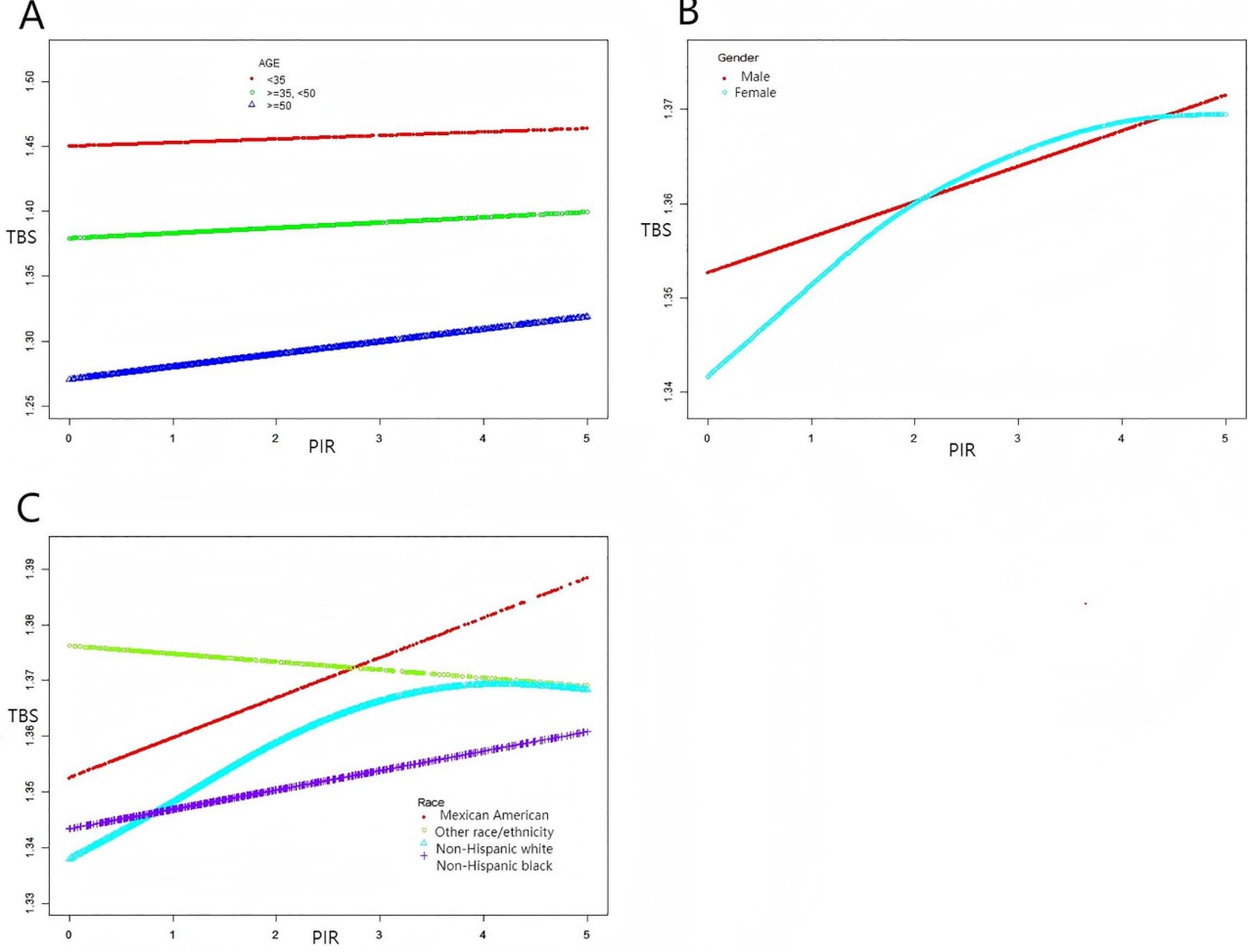

**Fig 3. The association between family PIR and total TBS stratified by age (A), gender (B) and race/ethnicity (C).** This model adjusted for all variables except family PIR, total TBS, and corresponding stratified variables.

Smoking has long been identified as a risk factor for osteoporosis [28]. A cross-sectional study including 10,564 participants showed a nonlinear and positive relationship between higher serum cotinine levels (≥3 ng/ml) and the prevalence of osteoporosis [29]. Additionally, individuals in lower income jobs are more likely to carry a higher risk of occupational injury and may have workplace exposure to environmental risk factors, including hazardous wastes and toxic chemical pollution (i.e., lead), as they are more likely to have industrial jobs [30]. This study found that serum cadmium and lead levels were significantly elevated in the low PIR group (Table 1). Heavy metal exposure may inhibit bone formation by downregulating runt-related transcription factor 2 (RUNX2) expression [31] and activate the nuclear factor kappa-B (NF-κB) pathway to promote bone resorption [32,33], thereby exacerbating the deterioration of bone microstructure. A large cross-sectional study revealed a dose-dependent relationship between TBS and bone cadmium (B-Cd) for both males and females [34].

Moreover, working individuals, citizens, and those with higher economic status were more likely to have better access to healthcare through health insurance. High-income individuals can afford the treatment and management costs of chronic diseases such as diabetes, hypertension, and osteoporosis, and take timely preventive measures and treatment [35]. In contrast, persons who lack insurance receive less medical care, including screening and treatment, and may

receive poorer-quality care [36]. Compared to those with less education, well-educated respondents are more likely to be employed, earn higher incomes, and experience less economic hardship. They also tend to have extensive health knowledge, contributing to a healthier lifestyle [37]. Overall, citizenship, working status, and economic status may serve as surrogate markers for the healthcare axis and a better lifestyle, revealing the relationship between family SES and total TBS from multiple perspectives. In summary, individuals with lower family SES may face challenges in areas such as nutritional intake, lifestyle, work-life environment, chronic disease management, and health beliefs, which can lead to osteoporosis. To clarify the relationship between family SES and total TBS, it is necessary to conduct more empirical research to understand how these factors interact and influence bone health.

Our study still has several limitations. First of all, due to its cross-sectional nature, we were unable to determine the temporal relationship between PIR and TBS. Secondly, the included covariates may not be completely accurate, which can affect the accuracy of the results. Despite these shortcomings, there are some strengths in our study, such as the fact that it is founded on a great deal of representative data from NHANES. We have demonstrated a positive association between family PIR and total TBS status. Additionally, we were able to undertake subgroup analyses of family PIR and total TBS across age, gender, and ethnicity, which has diagnostic value in predicting the presence of osteoporosis in a patient.

## Conclusion

Our results suggest that family PIR positively correlated with total TBS among U.S adults. Differences of PIR in the population should be considered in the diagnosis and treatment of osteoporosis.

## Acknowledgments

The author thanks the NHANES staff and participants for their valuable contributions.

## Author contributions

**Conceptualization:** Yuan Zhang, Tingxiao Zhao, Qice Sun.

**Writing – original draft:** Yuan Zhang, Tingxiao Zhao, Yanlei Li.

**Writing – review & editing:** Kai Chen, Qice Sun.

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
