## [Decision Letter · Decision Letter 0]

23 Jun 2025

Dear Dr. Sun,

Thank you for submitting your manuscript to PLOS ONE. After careful consideration, we feel that it has merit but does not fully meet PLOS ONE’s publication criteria as it currently stands. Therefore, we invite you to submit a revised version of the manuscript that addresses the points raised during the review process.

We look forward to receiving your revised manuscript.

Kind regards,

Khabir Ahmad, MBBS, MSc, Ph.D.

Academic Editor

PLOS ONE

Journal Requirements:

https://www.frontiersin.org/journals/endocrinology/articles/10.3389/fendo.2022.857110/full

https://pubmed.ncbi.nlm.nih.gov/36626464/

https://link.springer.com/article/10.1007/s00198-005-1917-1?

In your revision ensure you cite all your sources (including your own works), and quote or rephrase any duplicated text outside the methods section. Further consideration is dependent on these concerns being addressed.

3. Please note that your Data Availability Statement is currently missing the DOI/accession number of each dataset or a direct link to access each database. If your manuscript is accepted for publication, you will be asked to provide these details on a very short timeline. We therefore suggest that you provide this information now, though we will not hold up the peer review process if you are unable.

**Additional Editor Comments:**

Ensure all abbreviations are defined at first use.

Introduction

Change “This study addresses the gap in ...general population”

to

"This study aims to address the gap in ... general population.”

Fig 1 should be part of the methods. Remove arrows from figure. Use neutral connectors instead.

Methods

Provide the rationale for the selection of confounders and the choice statistical methods, especially for stratified and interaction analyses.

Results

Table 3 is too crowded; please divide it into two separate tables to improve clarity.

Discussion

Discuss the clinical significance of the small observed effect size?

Discuss possible mechanisms linking SES and TBS.

Explicitly mention study limitations,....use "temporal" instead of "causal"

Ethics

Add a brief statement clarifying that no ethics approval was needed due to the use of publicly available, de-identified NHANES data. Please check journal's guidelines

Reviewers' comments:

Reviewer's Responses to Questions

**Comments to the Author**

1. Is the manuscript technically sound, and do the data support the conclusions?

Reviewer #1: Yes

2. Has the statistical analysis been performed appropriately and rigorously?

Reviewer #1: Yes

3. Have the authors made all data underlying the findings in their manuscript fully available?

Reviewer #1: Yes

4. Is the manuscript presented in an intelligible fashion and written in standard English?

Reviewer #1: Yes

Reviewer #1: I don’t have a lot of concerns about this study – large sample size and an unsurprising finding demonstrating a relationship between SES and TBS. There are these three things I suggest the authors correct, including:

1. Introduction needs a hypothesis.

2. Introduction needs expanding to include literature about other studies have looked at the relationship between SES and osteoporosis/bone health. There are plenty of them and the readers will need to get a good picture of what has already been done in this space, before understanding that there is a need to look at TBS. In fact, there have been studies that found inconsistent relationships between SES and different measures of bone remodelling/BMD etc. It will be important to establish a good overview of the current state of the literature there.

3. In Figures 2 and 3, the font on the x and y axes and in the legend, is too small.

**Do you want your identity to be public for this peer review?** For information about this choice, including consent withdrawal, please see our Privacy Policy

Reviewer #1: No

---

## [Author Response · Author response to Decision Letter 1]

24 Jul 2025

Dear Editors and Reviewers:

Thank you for your letter and the valuable comments from the reviewers. Based on your constructive suggestions, we have revisited and revised the manuscript. These comments provide important guidance for improving the quality of the paper. All modifications and additions have been marked in blue font. We sincerely appreciate your positive evaluation and suggestions, and

hope that the revised manuscript will be accepted for publication in PLOS ONE.

Editor’s comments

Thank you for submitting your manuscript to PLOS ONE. After careful consideration, we feel that it has merit but does not fully meet PLOS ONE’s publication criteria as it currently stands. Therefore, we invite you to submit a revised version of the manuscript that addresses the points raised during the review process.

Answer: Thank you for your affirmation and recognition of our article. We are fully aware that the manuscript still has shortcomings, and your professional comments are extremely valuable for improving the quality of the paper. We have made comprehensive revisions based on your suggestions and look forward to your further review and approval.

Journal Requirements:

1.Please ensure that your manuscript meets PLOS ONE's style requirements, including those for file naming. The PLOS ONE style templates can be found at https://journals.plos.org/plosone/s/file?id=wjVg/PLOSOne_formatting_sample_main_body.pdf and

Answer: Thank you for your comments. This task has been completed according to your requirements.

2.We noticed you have some minor occurrence of overlapping text with the following previous publication(s), which needs to be addressed:

https://www.frontiersin.org/journals/endocrinology/articles/10.3389/fendo.2022.857110/full

https://pubmed.ncbi.nlm.nih.gov/36626464/

https://link.springer.com/article/10.1007/s00198-005-1917-1?

In your revision ensure you cite all your sources (including your own works), and quote or rephrase any duplicated text outside the methods section. Further consideration is dependent on these concerns being addressed.

Answer: Thank you for your comments. Your suggestions are very important for improving the quality of the manuscript. We sincerely apologize for the text overlap issue and have thoroughly revised the manuscript.

3.Please note that your Data Availability Statement is currently missing the DOI/accession number of each dataset or a direct link to access each database. If your manuscript is accepted for publication, you will be asked to provide these details on a very short timeline. We therefore suggest that you provide this information now, though we will not hold up the peer review process if you are unable.

Answer: Thank you for your comments. Your suggestions are very important for improving the quality of the manuscript. We have addressed your comments and completed the task as per your request.

Answer: Thank you for your comments. This task has been completed according to your requirements.

5.Please include a separate caption for each figure in your manuscript.

Answer: Thank you for your comments. This task has been completed according to your requirements.

Additional Editor Comments:

Ensure all abbreviations are defined at first use.

Answer: Thank you for your comments. This task has been completed according to your requirements.

Introduction

Change “This study addresses the gap in ...general population”

to "This study aims to address the gap in ... general population.”

Answer: Thank you for your comments. Your suggestions are very important for improving the quality of the manuscript. We have addressed your comments and completed the task as per your request.

Fig 1 should be part of the methods. Remove arrows from figure. Use neutral connectors instead.

Answer: Thank you for your comments. Your suggestions are very important for improving the quality of the manuscript. Arrows, as internationally recognized symbols in flowcharts, are widely used in academic journal flowcharts. They visually demonstrate the sequence and direction of screening steps, conforming to international guideline standards and readers' reading habits, thereby enhancing the readability and academic quality of diagrams.

Methods

Provide the rationale for the selection of confounders and the choice statistical methods, especially for stratified and interaction analyses.

Answer: We thank the reviewer for this comment. We have added a clear rationale for our selection of confounders and statistical methods in the Methods section.

Results

Table 3 is too crowded; please divide it into two separate tables to improve clarity.

Answer: Thank you for your comments. Your suggestions are very important for improving the quality of the manuscript. We have addressed your comments and completed the task as per your request.

Discussion

Discuss the clinical significance of the small observed effect size?

Answer: Thank you for your comments. We acknowledge the importance of addressing the clinical significance of the observed effect size. We have added a new paragraph in the Discussion section to specifically address this point.

Discuss possible mechanisms linking SES and TBS.

Answer: Thank you for your comments. Your suggestions are very important for improving the quality of the manuscript. We have addressed your comments and completed the task as per your request.

Explicitly mention study limitations,....use "temporal" instead of "causal"

Answer: Thank you for your comments. This task has been completed according to your requirements.

Ethics

Add a brief statement clarifying that no ethics approval was needed due to the use of publicly available, de-identified NHANES data. Please check journal's guidelines

Answer: Thank you for your comments. This task has been completed according to your requirements.

Comments to the Author

1. Is the manuscript technically sound, and do the data support the conclusions?

Reviewer #1: Yes

Answer: Thank you for your comments. We appreciate your positive remarks on our manuscript. We sincerely hope that our article will be accepted for publication.

2. Has the statistical analysis been performed appropriately and rigorously?

Reviewer #1: Yes

Answer: Thank you for your comments. We appreciate your positive remarks on our manuscript. We sincerely hope that our article will be accepted for publication.

3. Have the authors made all data underlying the findings in their manuscript fully available?

Reviewer #1: Yes

Answer: Thank you for your comments. We appreciate your positive remarks on our manuscript. We sincerely hope that our article will be accepted for publication.

4. Is the manuscript presented in an intelligible fashion and written in standard English?

Reviewer #1: Yes

Answer: Thank you for your comments. We appreciate your positive remarks on our manuscript. We sincerely hope that our article will be accepted for publication.

6.Review Comments to the Author

Answer: Thank you again for your positive and constructive comments and suggestions on our manuscript. We hope you will find our revised manuscript acceptable for publication.

Reviewer #1: I don’t have a lot of concerns about this study – large sample size and an unsurprising finding demonstrating a relationship between SES and TBS. There are these three things I suggest the authors correct, including:

 Answer: We feel great thanks for your professional review work on our article. As you are concerned, there are several problems that need to be addressed. According to your nice suggestions, we have made extensive corrections to our previous draft, the detailed corrections are listed below.

1. Introduction needs a hypothesis.

Answer: Thank you for your comments. Your suggestions are very important for improving the quality of the manuscript. We have addressed your comments and completed the task as per your request.

2. Introduction needs expanding to include literature about other studies have looked at the relationship between SES and osteoporosis/bone health. There are plenty of them and the readers will need to get a good picture of what has already been done in this space, before understanding that there is a need to look at TBS. In fact, there have been studies that found inconsistent relationships between SES and different measures of bone remodelling/BMD etc. It will be important to establish a good overview of the current state of the literature there.

Answer: Thank you for your comments. Your suggestions are very important for improving the quality of the manuscript. We have addressed your comments and completed the task as per your request.

3. In Figures 2 and 3, the font on the x and y axes and in the legend, is too small.

Answer: Thank you for your comments. This task has been completed according to your requirements.

6. PLOS authors have the option to publish the peer review history of their article (what does this mean?). If published, this will include your full peer review and any attached files.

Do you want your identity to be public for this peer review? For information about this choice, including consent withdrawal, please see our Privacy Policy.

Reviewer #1: No

We sincerely thank the editor and all reviewers for their valuable feedback that we have used to improve the quality of our manuscript. We hope our revisions meet your approval and contribute meaningful insights to the field. We welcome any further comments or suggestions.

Yours sincerely

Qice Sun

July 15, 2025

---

## [Decision Letter · Decision Letter 1]

8 Oct 2025

Dear Dr. Sun,

We look forward to receiving your revised manuscript.

Kind regards,

Jianhong Zhou

Staff Editor

PLOS ONE

Journal Requirements:

Reviewers' comments:

Reviewer's Responses to Questions

**Comments to the Author**

Reviewer #1: All comments have been addressed

2. Is the manuscript technically sound, and do the data support the conclusions?

Reviewer #1: Yes

3. Has the statistical analysis been performed appropriately and rigorously?

Reviewer #1: Yes

4. Have the authors made all data underlying the findings in their manuscript fully available?

Reviewer #1: Yes

5. Is the manuscript presented in an intelligible fashion and written in standard English?

Reviewer #1: Yes

Reviewer #1: This is a much-improved manuscript. I have made some very minor comments in text on the writing mechanics here and there - please see file attached. I also suggest to rephrase the elements of the text that refer to the 'racial' analysis - it would be more culturally sensitive to use terms such as ethnicity or ancestry rather than race. Same goes for the use of 'blacks' 'whites' etc. - please consider using terms such as African-Americans, Caucasians etc.

**Do you want your identity to be public for this peer review?** For information about this choice, including consent withdrawal, please see our Privacy Policy

Reviewer #1: No

---

## [Author Response · Author response to Decision Letter 2]

4 Nov 2025

To:

The Editorial Office

PLOS ONE

Subject: Revised Submission of Manuscript PONE-D-24-55944R1

Title: Unveiling the Wealth-Bone Connection: How Socioeconomic Status Influences Trabecular Bone Health

Dear Editor,

We sincerely thank you and the reviewer for your constructive and thoughtful feedback on our manuscript. We are pleased that Reviewer #1 found our previous revisions satisfactory and considered the manuscript much improved.

In accordance with the reviewer’s remaining suggestions, we have made several minor textual revisions to improve clarity and cultural sensitivity. Specifically, we replaced terms such as race, blacks, and whites with more culturally appropriate and scientifically precise expressions including ethnicity, ancestry, African Americans, and Caucasians. These changes have been carefully applied throughout the Methods, Results, and Discussion sections.

We have also reviewed the reference list for completeness and accuracy, ensuring compliance with the PLOS ONE reference and data availability policies.

A detailed, point-by-point rebuttal addressing the reviewer’s comments is included as a separate document, and all textual changes have been highlighted in the revised manuscript for ease of review.

We thank the reviewers and editorial team again for their time and effort in improving the quality of our work. We believe that the revised version of our manuscript now meets the journal’s publication standards, and we respectfully submit it for your further consideration.

Sincerely,

Qice Sun

The Second Affiliated Hospital of Zhejiang Chinese Medical University

xinhuasunqice@163.com

Response to Review Comments

Manuscript Title: Unveiling the Wealth-Bone Connection: How Socioeconomic Status Influences Trabecular Bone Health

Manuscript ID: PONE-D-24-55944R1

Dear Editor,

We sincerely thank you and the reviewer for the positive evaluation of our revised manuscript. We are pleased that Reviewer #1 found our responses satisfactory and considered the manuscript much improved.

Below, we provide our responses to the minor remaining suggestions.

Editorial Requirements

We have also reviewed the reference list to ensure it is accurate and that no retracted works are cited. All data remain publicly available in accordance with the PLOS Data Policy.

Journal Requirements:

Response Thank you for your attention. The reviewer's comments did not include suggestions for citing published literature.

Response Thank you for your attention. We carefully checked the list of references to ensure that the content was complete and accurate, and that there were no papers that had been retracted.

Reviewers' comments:

Reviewer's Responses to Questions

Comments to the Author

1. If the authors have adequately addressed your comments raised in a previous round of review and you feel that this manuscript is now acceptable for publication, you may indicate that here to bypass the “Comments to the Author” section, enter your conflict of interest statement in the “Confidential to Editor” section, and submit your "Accept" recommendation.

Reviewer #1: All comments have been addressed

Response: Thank you for your attention.

2. Is the manuscript technically sound, and do the data support the conclusions?

Reviewer #1: Yes

Response: Thank you for your attention.

3. Has the statistical analysis been performed appropriately and rigorously?

Reviewer #1: Yes

Response: Thank you for your attention.

4. Have the authors made all data underlying the findings in their manuscript fully available?

Reviewer #1: Yes

Response: Thank you for your attention.

5. Is the manuscript presented in an intelligible fashion and written in standard English?

Reviewer #1: Yes

Response: Thank you for your attention.

6. Review Comments to the Author

Reviewer #1: This is a much-improved manuscript. I have made some very minor comments in text on the writing mechanics here and there - please see file attached. I also suggest to rephrase the elements of the text that refer to the 'racial' analysis - it would be more culturally sensitive to use terms such as ethnicity or ancestry rather than race. Same goes for the use of 'blacks' 'whites' etc. - please consider using terms such as African-Americans, Caucasians etc.

Response: We thank the reviewer for this valuable suggestion. Following this advice, we have carefully revised the manuscript to replace all terms referring to “race” with more culturally appropriate and scientifically precise terminology. Specifically:

“race” has been replaced by “ethnicity” or “ancestry,” depending on context.

“blacks” and “whites” have been replaced with “African Americans” and “Caucasians,” respectively.

All corresponding sentences have been reviewed to ensure consistent and respectful language throughout the manuscript.

These changes have been made in the Methods, Results, and Discussion sections accordingly (highlighted in the revised version).

7. PLOS authors have the option to publish the peer review history of their article (what does this mean?). If published, this will include your full peer review and any attached files.

Do you want your identity to be public for this peer review? For information about this choice, including consent withdrawal, please see our Privacy Policy.

Reviewer #1: No

Response: Thank you for your attention.

---

## [Decision Letter · Decision Letter 2]

23 Nov 2025

Unveiling the Wealth-Bone Connection: How Socioeconomic Status Influences Trabecular Bone Health

PONE-D-24-55944R2

Dear Dr. Sun,

We’re pleased to inform you that your manuscript has been judged scientifically suitable for publication and will be formally accepted for publication once it meets all outstanding technical requirements.

Kind regards,

Mukhtiar Baig, Ph.D.

Academic Editor

PLOS ONE

Reviewers' comments:

Reviewer's Responses to Questions

**Comments to the Author**

Reviewer #1: All comments have been addressed

2. Is the manuscript technically sound, and do the data support the conclusions?

Reviewer #1: Yes

3. Has the statistical analysis been performed appropriately and rigorously?

Reviewer #1: Yes

4. Have the authors made all data underlying the findings in their manuscript fully available?

Reviewer #1: Yes

5. Is the manuscript presented in an intelligible fashion and written in standard English?

Reviewer #1: Yes

Reviewer #1: Thank you for making the final minor corrections, the manuscript is now ready for publication, in my opinion.

**Do you want your identity to be public for this peer review?** For information about this choice, including consent withdrawal, please see our Privacy Policy

Reviewer #1: No

---

## [Editor Report · Acceptance letter]

PONE-D-24-55944R2

PLOS ONE

Dear Dr. Sun,

I'm pleased to inform you that your manuscript has been deemed suitable for publication in PLOS ONE. Congratulations! Your manuscript is now being handed over to our production team.

Kind regards,

on behalf of

Professor Mukhtiar Baig

Academic Editor

PLOS ONE